# Implementation status of national tuberculosis infection control guidelines in Bangladeshi hospitals

**Arifa Nazneen**[1]◉*, **Sayeeda Tarannum**[1]◉, **Kamal Ibne Amin Chowdhury**[1], **Mohammad Tauhidul Islam**[1‡], **S. M. Hasibul Islam**[1‡], **Shahriar Ahmed**[1], **Sayera Banu**[1], **Md Saiful Islam**[1,2]

**1** Programme for Emerging Infections, Infectious Disease Division, International Centre for Diarrhoeal Disease Research, Bangladesh (icddr,b), Mohakhali, Dhaka, Bangladesh, **2** School of Public Health and Community Medicine, Faculty of Medicine, University of New South Wales, Kensington, Australia

◉ These authors contributed equally to this work.
‡ These authors also contributed equally to this work.
* arifa.nazneen@icddrb.org

**Data Availability Statement:** Data are available from the data repository committee at icddr,b. The dataset underlying the findings described in the paper cannot be shared publicly due to ethical

## Abstract

In response to the World Health Organization (WHO) recommendation to reduce healthcare workers' (HCWs') exposure to tuberculosis (TB) in health settings, congregate settings, and households, the national TB control program of Bangladesh developed guidelines for TB infection prevention and control (IPC) in 2011. This study aimed to assess the implementation of the TB IPC healthcare measures in health settings in Bangladesh. Between February and June 2018, we conducted a mixed-method study at 11 health settings. The team conducted 59 key-informant interviews with HCWs to understand the status of and barriers impeding the implementation of the TB IPC guidelines. The team also performed a facility assessment survey and examined TB IPC practices. Most HCWs were unaware of the national TB IPC guidelines. There were no TB IPC plans or committees at the health settings. Further, a presumptive pulmonary TB patient triage checklist was absent in all health settings. However, during facility assessment, we observed patient triaging and separation in the TB specialty hospitals. Routine cough-etiquette advice was provided to the TB patients mentioned during the key-informant interviews, which was consistent with findings from the survey. This study identified poor implementation of TB IPC measures in health settings. Limited knowledge of the guidelines resulted in poor implementation of the recommendations. Interventions focusing on the dissemination of the TB IPC guidelines to HCWs along with regular training may improve compliance. Such initiatives should be taken by hospital senior leadership as well as national policy makers.

## Introduction

Tuberculosis (TB) is an infectious disease caused by *Mycobacterium tuberculosis*, causing the highest number of deaths as a single infectious agent globally [1]. In 2019, 10 million people were infected with TB globally; 79% were in the 30 high-burden countries, and 1.2 million people died from TB [1]. Bangladesh is one of the 30 high TB-burden countries and accounts for

restrictions related to protecting study participants' privacy and icddr,b's data access policy (https://www.icddrb.org/policies). icddr,b has a data repository maintains by the research administration. A copy of the complete dataset (anonymized and decoded) of this study will remain at the data repository. Interested researchers may contact Ms. Armana Ahmed, head of research administration (aahmed@icddrb.org), for approval and data access.

**Funding:** Md. Saiful Islam, senior author of this manuscript received the grant from USAID. The grant number was AID- 388-A-17-00006. This research activity was made possible by the generous support of the american people through the United States Agency for International Development (USAID).Funding source website: https://www.usaid.gov/bangladesh. icddr,b acknowledges with gratitude the commitment of the USAID to its research efforts. The funding source had no role in study design, data collection and analysis, decision to publish, or preparation of the manuscript.

**Competing interests:** The authors have declared that no competing interest exist.

3.6% of the global total. The estimated incidence of TB per 100,000 is 221 in Bangladesh, with a mortality rate of 24 per 100,000 population [1]. Approximately 80% of all TB cases in Bangladesh are pulmonary TB [2].

The Global TB Report 2020 estimated that 0.7% of new cases and 11% of previously treated cases are found to be positive for multidrug-resistant TB (MDR-TB), which has an incidence rate of 2.0 per 100,000 population in Bangladesh [1].

The Bangladesh national guidelines and operational manual for TB control recommend treating TB patients in a TB hospital or Directly Observed Treatment Short-course (DOTS) clinic [3]. For TB-patient treatment, DOTS therapy is considered the most effective and sustainable part of the National Tuberculosis Control Program (NTP). In hospitals, the guidelines recommend the enrollment and hospitalization of a drug-resistant TB patient or TB patient with co-morbidity in a designated TB or MDR-TB ward. Due to the high number of patients, limited number of beds, lengthy treatment procedure, and lack of patient monitoring mechanisms, the government of Bangladesh also initiated community-based programmatic management of drug-resistant TB [4]. The community-based programmatic management of drug-resistant TB guidelines recommend the admission of drug-resistant TB patients in chest disease hospitals for a minimum of four weeks or until two consecutive sputum smear microscopies become negative one week apart before sending them to the community.

In middle- and low-income high TB-burden countries, all healthcare workers (HCWs) are at risk of TB exposure due to the presence of presumptive and or confirmed TB patients in the hospital [5–7]. Public tertiary care hospitals often lack basic infection prevention and control (IPC) measures that make HCWs more vulnerable [8]. In low-and middle-income countries, the pooled prevalence of latent TB infection among HCWs was 47%, whereas in Bangladesh it was 54% [6, 9, 10].

The poor implementation of TB IPC measures may influence both latent TB infection and active disease among HCWs [11]. In low-income, high TB-burden settings, the factors responsible for the poor implementation of TB IPC measures were as follows: the absence of policies, poor knowledge, heavy workload, and a lack of training [12–15]. The lack of resource availability, gaps in behavioral motivation, lack of proper knowledge and training, and delegated leadership were also reported as major factors impeding IPC implementation [13, 16].

TB IPC has been prioritized in the WHO's updated Stop TB Strategy. Based on the WHO's 2009 policy on TB infection control in healthcare facilities, congregate settings, and households, the NTP of the government of Bangladesh developed TB IPC guidelines (http://etoolkits.dghs.gov.bd/sites/default/files/national_guidelines_for_tuberculosis_infection_control.pdf) in 2011 as a part of health system strengthening [17, 18]. The guidelines comprised a hierarchy of control measures that included managerial activities to strengthen coordination in the implementation of appropriate TB infection control measures; administrative controls to reduce the generation of aerosols and, thereby, the exposure to droplet nuclei; environmental controls to reduce concentrations of infectious particles; and personal protective measures to reduce inhalation and exhalation of infectious particles. To assist the NTP in implementing the TB IPC guidelines, it is important to understand the circumstances under which the hospitals have been implementing the guidelines since 2011. Therefore, this study aimed to assess the status of and barriers impeding the implementation of TB IPC measures in TB specialty hospitals and tertiary care hospitals in Bangladesh.

## Materials and methods

### Study sites

A field team of five members, consisting of an epidemiologist (one), social scientists (two), a physicist (one), and a medical technician (one), conducted this study in 11 health settings:

eight TB specialty hospitals (seven public and one private) and three tertiary care hospitals (two public and one private) in Bangladesh. The rationale for selecting these hospitals was based on the fact that TB specialty hospitals admit and treat TB patients on a regular basis, whereas tertiary care hospitals admit presumptive TB patients until diagnosis and subsequently refer confirmed TB patients to either DOTS clinics or TB specialty hospitals. These hospitals also serve the largest number of TB patients in the country. Based on our prior experience working in Bangladeshi hospitals, TB patient management and the implementation of TB IPC are likely to vary between government (public) and non-government (private) hospitals. Therefore, we also included private hospitals in our study.

## Study design and data collection

This was a mixed-method study. We used both qualitative and quantitative data collection tools that included key informant interviews (KIIs), observation, and a facility assessment checklist. The field team consisted of four males and one female researcher, trained in social science research with approximately five years of TB-related research experience, who collected the data. The field team had prior working relations with the study facility management teams, and this helped to build good rapport with the participants. The field team sought written permission from all the facilities before the data collection commenced. The TB specialty hospitals were situated in Dhaka, Rajshahi, Sylhet, Barishal, Chittagong, Khulna, Mymensingh, and Pabna, and the tertiary care hospitals were situated in Rajshahi, Barishal, and Kishoreganj. Between February and June 2018, the team conducted 59 unstructured KIIs with hospital directors [10], heads of medicine units (five), senior physicians (eight) and junior physicians (five) of inpatient and outpatient departments, laboratory personnel [19], and nursing supervisors [11] and administrative worker (one). The participants were selected purposively, and the interviewer approached the respondents face to face. After obtaining informed written consent, three researchers trained in social science with several years of experience in qualitative research conducted the KIIs in the Bengali language. Through KIIs, the team investigated the presence of a TB IPC committee or plan, surveillance and assessment of TB among HCWs, staff training, monitoring and evaluation of TB IPC, advocacy or communications for TB IPC implementation, triage and separation of TB patients, cough etiquette, and personal protective measures using respirators. All the interviews were audio-recorded, and the mean duration of the interviews was 42 min. Using an open-ended interview guide, the field team conducted the interviews. The time and venue for the interviews were selected based on the respondents' preferences. Each day, after data collection, the field team convened and discussed the interview findings. The team continued interviewing participants until data saturation was achieved, and no new data were obtained from additional interviews. We did not conduct any repeat interviews; however, a few of the respondents were re-engaged to clarify any findings from the interviews. Based on the findings, a report was prepared and shared with all participant hospitals for review and approval. Using a facility assessment tool, the team documented the presence or absence of the following: a TB IPC coordination committee or plan, TB surveillance among HCWs, training, triage, separation/cohorting of patients with pulmonary TB, cough etiquette, ventilation, fans, ultraviolet germicidal irradiation (UVGI), and respirators available for staff and fit tests or fit checks. A team of three field researchers conducted a total of 88 h (eight hours per facility) of structured observation. The tertiary care hospitals lacked a separate ward for TB patients. Presumptive pulmonary TB patients were admitted to the adult medicine wards with other general patients. Therefore, we conducted observations in the adult medicine wards of tertiary care hospitals. During observation, the team documented the number of functional fans, UVGI lights, doors and windows (including how many of them were

open), fanlights, and exhaust fans as well as the presence of air conditioning, use of N95 respirators among ward occupants, use of surgical/cloth masks, use of gloves, observance of cough etiquette, use of a triage checklist, separation of presumptive pulmonary TB patients, presence of posters, and presence of signs for restricted areas or any directional sign. The field team also looked for IPC posters in the outdoor, emergency, waiting, and entrance areas of the facilities.

### Data analysis

The team transcribed all audio recorded KIIs verbatim and reviewed each transcript at least twice. Three team members who were involved in data collection coded the data. The first authors developed a code list along with code definitions. The team then reviewed the transcriptions line by line, coded the data, and summarized the data under emerging and predefined themes based on the research questions that were aligned with the four broader TB IPC measures. When disagreements occurred, the team discussed the codes and their definitions with the senior author (MSI) to reach a consensus, and thus, intercoder agreement was achieved. The team also reviewed the facility survey data and extracted the frequency of the activities into a spreadsheet, and a descriptive analysis was performed. The facility assessment tool was adopted from the national tuberculosis infection control guidelines [17].

### Ethics

The icddr,b IRB has two separate committees: Research Review Committee (RRC) and Ethical Review Committee (ERC). The icddr.b's RRC reviewed the study protocol to ensure the scientific rigour and the validity of study design and data collection tools. The ERC reviewed the protocol from the perspective of human subject's research in Bangladesh. IRB approval number: PR#12067. The field team obtained written, informed consent from the study participants. This study protocol was also reviewed and approved by the NTP under the Ministry of Health and Family Welfare, Government of Bangladesh.

## Results

A total of 59 HCWs participated in the study; 28 were physicians, 11 nurses, 19 laboratory personnel, and one a project director. The project director was involved in the overall monitoring and supervision of the TB project activities as well as TB patient management in the hospital. The project director and 10 hospital directors were predominantly involved in administrative activities and implementation of policies recommended by the Ministry of Health and other partners. Physicians and nurses worked directly with TB patients, and lab workers were regularly exposed to patients and their specimens. The mean age of the participants was 45 years, with a mean job duration of 10 years. Thirty-four percent of the participants were men. The TB specialty hospitals were providing medical services mainly to patients who were critically ill with TB or had MDR-TB. In 2017, 818 and 385 confirmed pulmonary TB patients were cared for in TB specialty and tertiary care hospitals, respectively.

### Implementation of managerial control

As regards managerial control, we looked for an infection control coordination body, TB-IPC guidelines, and training on TB IPC. We also looked for monitoring and evaluation, operational research, advocacy communications, and social mobilization activities as well as surveillance and assessment of HCWs.

The key informants mentioned that there was no infection control coordinating body or person responsible for TB IPC in the hospitals, and similar findings were also observed in

**Table 1. Tuberculosis infection prevention and control practices and challenges in 11 study facilities, 2018.**

| TB-IPC measures | Reason for non-adherence/scope of improvement |
|---|---|
| **Managerial activities** | |
| Coordinating body or responsible person in place | Need instructions from the ministry and comprehensive plan |
| TB infection control plan (written) | Need instructions from the ministry |
| Surveillance among HCWs | Lack of assigned manpower, need proper guidelines and awareness |
| Training on infection control | Need instructions from higher authority at national level |
| Advocacy, communication, and social mobilization | Lack of assigned manpower and need for proper guidelines |
| Monitoring and evaluation | Lack of assigned manpower and heavy workload |
| Operational research | Need instructions from the ministry, budget, and comprehensive plan |
| **Administrative controls** | |
| Triage | Workload, no triage checklist, need awareness |
| Separation | Lack of dedicated waiting areas, patient overload |
| Cough-etiquette education | Nurses provide instructions |
| Expedient service delivery | Lack of resources and assigned manpower |
| Prevention care/risk allowance | No risk allowance is allocated for HCWs in tertiary care hospitals |
| **Environmental Controls** | |
| Natural and mechanical ventilation | Need proper monitoring |
| Fans | Proper monitoring needed |
| Ultraviolet germicidal irradiation | Instruction on UVGI use and monitoring needed |
| **Personal Protective Equipment** | |
| Respirators available for staff | Supply should be increased to all health settings |

facility assessment data. None of the respondents received training on biosafety and biosecurity, except one. Monitoring of laboratory equipment functionality was operational in only four TB specialty hospitals. Laboratory personnel in TB specialty hospitals tended to wear N95 respirators and gloves to ensure TB IPC when closely handling TB specimens more than the HCWs dealing with patients in the wards did. In addition, none of the study hospitals conducted any operational research on TB. Further, there were no advocacy, communication, or social mobilization activities available in the study settings (Table 1). There was also no TB surveillance system for HCWs.

**Challenges in implementing managerial control activities.** All participating physicians mentioned that they never received any verbal or written instructions to form a committee for TB IPC in the hospital. A consultant physician from a TB specialty hospital stated the following:

*Instruction should come from a higher authority. If the ministry provides instruction, the hospital authorities will work on it. The ministers, secretaries, and directors should promote infection control; they are the main managers.*

When asked about challenges hindering the formation of an IPC committee, the respondents mentioned that they had not received a detailed plan, instructions, and budget from the authorities responsible for TB IPC. A director of a TB specialty hospital claimed:

*How would we do that? Does it not require money? Who will give me the money? (Pointing a tube light) this light is not working along with the switch board; it will need a socket and a switch to repair. Where would I get the money to buy these?*

From the interviews, we discovered that the hospitals lacked surveillance and assessment of TB infection and disease among HCWs. The physicians mentioned a lack of assigned persons and proper guidelines as barriers to surveillance and assessment. The respondents mentioned that if anyone complained of illness, they could do the TB tests free of cost. HCWs are required to have chest X-rays taken annually for their annual confidential report; however, some of the participants reportedly completed the annual confidential forms without having their chest X-rays taken to avoid reporting their disease status. None of the study respondents were aware of any written plan on TB IPC. A senior physician from a tertiary care hospital said:

*Each hospital should have a TB program, and the hospital should instantly form an infection control committee. The committee will be responsible for surveillance implementation and periodically inform the director about their activities. The Director will send a report to the central authority. If the chain could be established, everything will be done smoothly. However, we have not received any papers from the government yet.*

None of the study participants had received any training on TB IPC. However, they reportedly received training on MDR-TB treatment regimens. A nurse from a TB specialty hospital said:

*I have never received any training on infection control, and I have never heard about this type of training. My colleagues who have worked here for many years have also never received any training on infection control.*

Regarding operational research, the respondents repeatedly mentioned that research requires a budget, planning, and manpower. One of the study settings (private TB specialty hospital) conducted operational research on TB control. Almost all the respondents mentioned that they did not receive any budget or instructions to conduct operational research. A physician from a tertiary care hospital reported the following:

*It should be implemented by the NTP, and the ministry could send letters to hospital directors to initiate research activity.*

## Implementation of administrative control measures

In terms of administrative control, we looked for triage, a triage checklist, and isolation or separation of suspected TB patients. We also looked for cough etiquette and extracted information regarding the availability of a prevention and care package as well as risk allowances for HCWs.

All the respondents reported that they would always try to maintain a triage for presumptive pulmonary TB patients and cohort or isolate the pulmonary TB patients. They also mentioned that the nurses often conducted patient counseling on cough etiquette; however, they were not consistent due to time constraints and workload. The respondents stated that there was no prevention or healthcare package available for HCWs in the study hospitals; for example, the facilities did not have any periodic TB screening system, any workplace policy, disease surveillance, and notification system in place.

During the facility assessment survey, we observed the triaging of presumptive pulmonary TB patients in only two TB specialty settings (Table 2). We noted that these hospitals were caring for sputum-positive and MDR-TB patients separately, as they had separate wards for drug-susceptible and drug-resistant TB patients. In the tertiary care teaching hospitals, there was no

**Table 2. TB-IPC assessments at the study settings, based on facility assessment.**

| Type of facilities | Posters | Triage-observed | Triage-checklist present | Cough etiquette | Segregation of presumptive TB patients | Natural ventilation | Mechanical ventilation | N95 respirators for HCWs | | Patient with a surgical mask | | Exhaust fan | | UVGI | |
|---|---|---|---|---|---|---|---|---|---|---|---|---|---|---|---|
| | | | | | | | | Availability | HCWs using N95 | Self-provided | Hospital provided | Functioning | Non-functioning | Functioning | Non-functioning |
| TB specialty hospitals | ✓ | ✓ | X | ✓ | ✓ | ✓ | ✓ | ✓ | ✓ | ✓ | X | 6 | 0 | 9 | 3 |
| | X | X | X | ✓ | ✓ | ✓ | ✓ | ✓ | ✓ | X | X | 0 | 8 | 3 | 4 |
| CDH | X | X | X | ✓ | ✓ | ✓ | ✓ | ✓ | ✓ | ✓ | X | 1 | 3 | 0 | 1 |
| CDH | X | X | X | ✓ | X | ✓ | ✓ | ✓ | ✓ | X | X | 0 | 3 | 0 | 0 |
| CH | X | X | X | ✓ | ✓ | ✓ | X | ✓ | ✓ | X | X | 0 | 0 | 0 | 10 |
| CDH | X | X | X | ✓ | ✓ | ✓ | ✓ | ✓ | ✓ | ✓ | X | 3 | 1 | 8 | 0 |
| | ✓ | X | X | ✓ | ✓ | ✓ | ✓ | ✓ | ✓ | X | ✓ | 10 | 0 | 2 | 3 |
| | X | X | X | X | ✓ | ✓ | X | ✓ | X | X | X | 0 | 0 | 0 | 2 |
| Tertiary care hospitals | X | ✓ | X | X | ✓ | ✓ | ✓ | ✓ | ✓ | X | X | 3 | 0 | 0 | 0 |
| | ✓ | X | X | ✓ | X | X | ✓ | X | X | X | ✓ | 0 | 0 | 0 | 0 |

✓ = Available and X = not available.

separation or isolation of presumptive pulmonary TB patients, and interview participants mentioned that limited space was a barrier to triage and separation (Table 2). Three public and one private TB specialty hospital collected sputum outdoors (outside buildings, in an open place). The three public TB specialty hospitals collected sputum in the outdoor corridor, whereas three tertiary care hospitals and one public TB specialty hospital collected sputum in a designated indoor area. All hospitals lacked clear signs, such as "restricted area" signs, directional signage, or hospital guidance signage to assist people and keep them away from restricted areas. Typically, the pathologists would not collect sputum samples themselves; rather, they trained laboratory assistants to collect sputum and concurrently enforce the maintenance of cough etiquette by patients (patients were instructed on cough etiquette before sputum induction procedures). The laboratory assistant safely disposed of the sputum cup and sticks in the dedicated waste disposal container. Hence, duty doctors or nurses were not involved in sputum collection.

**Challenges in implementing administrative controls.** Although all our respondents reportedly implemented triage for presumptive pulmonary TB patients, we were unable to find a triage checklist at any setting (Table 2). Four of the eight TB specialty hospitals had dedicated TB outpatient waiting areas. In hospitals that lacked dedicated waiting areas for TB patients, infectious and non-infectious patients shared the same waiting area. A consultant physician from a TB specialty hospital stated the following:

> *Is it possible to separate patients? For example, we have accommodation for 120 patients in the hospital: 60 are sputum positive, and 60 are sputum negative. Now, if 10 patients become sputum negative from the positive patients, where will I accommodate them? I am bound to keep them together with sputum positive patients.*

Participants claimed that whenever they identified a patient with a cough, they would advise the patient to cover their mouth with a handkerchief, napkin or a tissue. However, there was no supply of masks to patients in the study hospitals; only the private TB specialty hospital provided surgical masks to their patients (Table 2). Approximately all of our respondents mentioned that their hospitals lacked resources or logistics and manpower for expedient service delivery (Table 1). None of the hospitals had developed a workplace policy regarding TB IPC. Practically all participating respondents reported having no TB screening for HCWs. No health education on the signs and symptoms of TB was reported, and there was no case-notification system for TB among HCWs.

## Implementation of environmental control measures

With respect to environmental controls, we searched for ventilation (natural or mechanical), the presence of fans, and a UVGI device.

Findings from the KIIs and facility assessment revealed that the study settings had good natural ventilation. A few participants perceived that having a TB patient in the inpatient ward did not mean that the ward was full of TB bacteria. As there was good natural ventilation in the ward, one respondent believed that the fresh air removed the germs from the ward. They also believed that if someone was to stay in the inpatient ward for a short duration, TB infection would not be a threat.

Approximately half of the TB specialty hospitals had functioning exhaust fans and three hospitals had non-functioning ones. On the other hand, only half of the TB specialty hospitals had functioning UVGI devices. We found no UVGI device in any of the three tertiary-care hospitals (Table 2). We searched for UVGI lights outdoors as well as in the emergency department, medicine wards, and pathology and radiology departments. Informants

reported that the hospital authorities never received any verbal or written instructions on UVGI use. One nurse from a TB specialty hospital said that they preferred to switch UVGI on for an hour every night. Nurses from another TB specialty hospital claimed that they switched on the UVGI light twice a day for an hour each instance. Although the study hospitals had enough ceiling fans, their repair and replacement were reported to be challenging and time consuming. A physician from a TB specialty hospital reported the following:

*The government public works department is responsible for providing fans. We lacked some fans; last year, we raised a requisition for seven fans. The executive engineer was supposed to send us the fans, but we do not know whether we will get those fans or not.*

### Availability and use of personal protective equipment

In terms of personal protective equipment, we investigated the availability of N95 respirators and fit-testing training/fit checks. Participants from the TB specialty hospitals reported that they had a supply of N95 respirators for HCWs. However, the tertiary care hospitals had no N95 respirators available for staff.

A physician from a TB specialty hospital stated the following:

*We have to indent for (N95) respirators; for example, if we place requirements for 1200 respirators, we receive only 800 (N95 respirators).*

HCWs working in the MDR-TB ward and laboratories were found to be using N95 respirators during the observation session of the facility assessment survey. However, in the patient medicine ward of the tertiary care hospital, we observed no ward occupants using masks or N95 respirators.

A physician from a tertiary care hospital reported the following:

*We seldom wear N95 respirators, and I cannot remember when I last used N95 respirators while attending a pulmonary TB patient.*

None of the study participants reported receiving training on N95 respirator use or fit testing. Among the interviewed physicians from the tertiary care hospitals, five respondents had not even heard about N95 fit testing. A physician from a TB specialty hospital stated the following:

*What is it (N95 fit testing)? What does it mean? I have no idea about this.*

One respondent from the private TB specialty hospital was aware that his staff could have been wearing N95 respirators that were not well fitted. He suggested that the respirators might not have been properly sealed; hence, this issue needed scrutiny. He added that a fit test was paramount; however, he never received any training on fit testing. Moreover, they had no resources to conduct the fit test in their hospital.

### Discussion

This study identified poor implementation of the national TB IPC guidelines in TB specialty hospitals and tertiary-care general hospitals. The lack of hospital-level policies, HCW unawareness of the national TB IPC policy, and no supply of N95 respirators for tertiary-care general

hospitals render HCWs susceptible to airborne infection. Managerial activities were partially implemented due to lack of instructions from the authorized ministry, lack of manpower, limited budget, and lack of a detailed infection control plan. Lack of training in TB prevention among hospital staff may influence compliance, as found in other studies conducted in China and Nigeria [19–21].

The absence of a TB infection control policy at hospital level and lack of training among HCWs not only make HCWs unaware of the different TB IPC measures but also put them at risk of exposure that may further contribute to the non-adherence to other infection control healthcare measures in the country [16, 22]. Their interest in receiving training on TB IPC demonstrated that there was a demand for TB infection control training among HCWs. Ministries or development partners involved in training on TB infection control should extend their intervention to TB specialty and tertiary care hospitals. In this study, limited budget and manpower were identified as barriers to implementing managerial control activities, and this is consistent with findings from other low-income, high TB-burden countries, such as Nigeria and Uganda [21, 23, 24].

Administrative controls enable rapid identification, separation, and diagnosis, which reduce the contamination of air due to mycobacterium tuberculosis [24]. The challenges of having limited manpower dedicated to TB infection control, HCW workload, and lack of knowledge and awareness largely affected the implementation of TB IPC. These findings are consistent with prior studies conducted in low-income settings [21, 24, 25], and they suggest that there is a gap between policies and their implementation. Nine years have passed since the TB IPC policies were introduced in the country; however, the absence of a TB infection control committee or responsible person and impaired knowledge of TB transmissibility among HCWs indicated a lack of monitoring and evaluation of the policies. This revelation also raised concerns regarding who should be responsible for this monitoring and evaluation.

Although the national TB infection control policy recommended separating infectious and non-infectious TB patients, it was difficult for the tertiary care hospitals to screen presumptive pulmonary TB patients on arrival and separate them. In this study, the TB specialty hospitals predominantly maintained the segregation of general TB patients from multi-drug resistant TB patients, with very limited implementation of managerial and administrative control measures.

Environmental control aims to reduce the concentrations of infectious particles in the surroundings. The presence of large window fans, spacious doors, and windows allowed good natural ventilation in the study areas, suggesting that these areas were well ventilated. However, people often switch off ceiling fans and keep the windows closed to avoid cool air and the mosquito menace [26, 27].

The study identified adherence to N95 respirator use among HCWs serving in MDR-TB wards and in the laboratory, and similar findings were observed in a cross-sectional study among HCWs in Vietnam and India [28]. Although there was a supply of N95 respirators at the TB specialty hospital, they were not being used in wards with drug-susceptible TB patients. Moreover, the lack of facemask supply to TB patients created an opportunity for airborne contamination of wards with *Mycobacterium* tuberculosis. In addition, the HCWs from all hospitals cited a lack of training on N95 respirator use and fit testing. Studies in other high TB-burden countries showed that training on N95 respirator use and fit testing increased adherence to N95 respirator use [8, 29].

Most studies, including this one, have identified poor adherence to the implementation of TB IPC guidelines among HCWs. However, one study conducted in Brazil by Azeredo et al. noted that the implementation of infection control measures, such as having a TB infection-control plan, periodic monitoring of the TB IPC, and the training of HCWs on infection control can decrease TB incidence in healthcare settings [30].

This study has several limitations. Most of the key informants were senior HCWs. These senior management employees may not have free time to participate in TB IPC training; therefore, they remain unaware of the TB IPC activities. Second, this study was conducted in 11 hospitals that were not randomly selected and therefore, the findings may not be generalizable to all hospitals that provide care to TB patients. However, this study findings are consistent with prior reports and studies conducted in other government and non-government hospitals in Bangladesh [7, 10, 31].

To enable the health system to better implement the national TB IPC guidelines, the guidelines should be introduced in all health settings dealing with TB cases through participatory approaches. Regular, intense monitoring by the NTP's infection-control coordinating bodies can be maintained as far as triage and segregation of TB patients is concerned. The national TB infection control guidelines should be rolled out in chest-disease hospitals, clinics, and tertiary care hospitals. Frontline HCWs should be trained on different IPC measures outlined in the national TB infection control guidelines. Interventions focusing on the dissemination of the national TB IPC guidelines at facility level, engagement of managerial level HCWs in building an infection control committee, training on N95 respirators use and fit testing, and monitoring and evaluation of TB infection control measures should be implemented. Awareness programs in communities, schools, and at all levels of health settings can improve TB IPC practices.

In conclusion, this study discovered that there was no initiative to disseminate the national TB IPC guidelines to hospitals, resulting in poor adherence to TB IPC measures among hospital HCWs. The HCWs' willingness to comply with respiratory controls may facilitate the implementation of regular facemask/respirator supply, face seal, and fit testing. From discussions with key informants, it was evident that the health settings prioritized patient management over TB infection control; however, TB prevention through infection control also needs to be a priority in the management of TB patients. Establishing an infection control committee, providing training to HCWs, and monitoring and evaluating infection control activities have to be in place to effectively implement infection control. Relevant government authorities may improve TB infection control practices by addressing the scope of improvements.

## Supporting information

**S1 File. TB infection control measures to be implemented (Facility assessment tool).** (DOCX)

**S2 File. Consolidated criteria for reporting qualitative studies (COREQ).** (DOCX)

## Acknowledgments

We express our gratitude to all participants from the study facilities for their time, unconditional support, and guidance. We are thankful to the National Tuberculosis Control Program of the government of Bangladesh for their continuous support. We would also like to thank Editage (www.editage.com) for English language editing.

## Author Contributions

**Conceptualization:** Arifa Nazneen, Sayera Banu, Md Saiful Islam.

**Formal analysis:** Arifa Nazneen, Sayeeda Tarannum, Kamal Ibne Amin Chowdhury, Mohammad Tauhidul Islam, S. M. Hasibul Islam, Md Saiful Islam.

**Funding acquisition:** Sayera Banu, Md Saiful Islam.

**Investigation:** Sayeeda Tarannum, Kamal Ibne Amin Chowdhury, Mohammad Tauhidul Islam, S. M. Hasibul Islam.

**Methodology:** Arifa Nazneen, Sayeeda Tarannum, Kamal Ibne Amin Chowdhury, Md Saiful Islam.

**Project administration:** Arifa Nazneen, Shahriar Ahmed, Sayera Banu, Md Saiful Islam.

**Resources:** Arifa Nazneen, Sayera Banu, Md Saiful Islam.

**Supervision:** Arifa Nazneen, Kamal Ibne Amin Chowdhury, Sayera Banu, Md Saiful Islam.

**Validation:** Arifa Nazneen, Sayeeda Tarannum, Md Saiful Islam.

**Writing – original draft:** Arifa Nazneen, Sayeeda Tarannum.

**Writing – review & editing:** Arifa Nazneen, Sayeeda Tarannum, Kamal Ibne Amin Chowdhury, Mohammad Tauhidul Islam, S. M. Hasibul Islam, Shahriar Ahmed, Sayera Banu, Md Saiful Islam.

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
