## [Decision Letter · Decision Letter 0]

6 Nov 2020

PONE-D-20-22170

Implementation status of national tuberculosis infection control guidelines in Bangladeshi hospitals

PLOS ONE

Dear Dr. Nazneen,

Thank you for submitting your manuscript to PLOS ONE. After careful consideration, we feel that it has merit but does not fully meet PLOS ONE’s publication criteria as it currently stands. Therefore, we invite you to submit a revised version of the manuscript that addresses the points raised during the review process.

Please find the comments from three reviewers below. The reviewers have requested some more context and detail in the methodology, in order to ensure that the study is fully reproducible by another researcher. Please note that while the reviewers have suggested specific papers for the literature review, there is no requirement from the journal to include these specific papers. We would also recommend that you thoroughly copyedit your manuscript, as some grammatical errors remain. If you do not know anyone who can help you with this, you may consider working with a professional copyeditor.

We look forward to receiving your revised manuscript.

Kind regards,

Hanna Landenmark

Associate Editor

PLOS ONE

Journal Requirements:

3. Under data analysis, the team reviewed the transcripts multiple times. Please specify how many times the team reviewed and if intercoder agreement was assessed.

4. Under study design, you indicate that the study utilized both qualitative and quantitative data - which were entered into MS excel for descriptive analysis. However, there are no text or tables summarizing quantitative findings. If there are no quantitative findings to report, please delete the methods section to show that only qualitative data were collected.

5. As part of your revision, please complete and submit a copy of the COREQ Guidelines checklist, a document that aims to improve experimental reporting and reproducibility of qualitative studies for purposes of post-publication data analysis and reproducibility: https://www.equator-network.org/reporting-guidelines/coreq/. Please include your completed checklist as a Supporting Information file. Note that if your paper is accepted for publication, this checklist will be published as part of your article

6. We note that you have indicated that data from this study are available upon request. PLOS only allows data to be available upon request if there are legal or ethical restrictions on sharing data publicly. For information on unacceptable data access restrictions, please see http://journals.plos.org/plosone/s/data-availability#loc-unacceptable-data-access-restrictions.

Reviewers' comments:

Reviewer's Responses to Questions

**Comments to the Author**

1. Is the manuscript technically sound, and do the data support the conclusions?

Reviewer #1: Yes

Reviewer #2: Yes

Reviewer #3: Partly

2. Has the statistical analysis been performed appropriately and rigorously? 

Reviewer #1: Yes

Reviewer #2: Yes

Reviewer #3: Yes

3. Have the authors made all data underlying the findings in their manuscript fully available?

Reviewer #1: No

Reviewer #2: No

Reviewer #3: Yes

4. Is the manuscript presented in an intelligible fashion and written in standard English?

Reviewer #1: Yes

Reviewer #2: Yes

Reviewer #3: No

5. Review Comments to the Author

Reviewer #1: In this article the authors aimed to assess the implementation of the TB IPC healthcare measures in health settings, Bangladesh. This is an important and interesting study.

They identified poor implementation of TB IPC measures in the study settings. In the ‘Discussion’ section, the authors cited other studies that found the same problems in implementation. However, it is important to cite studies that have been successful in implementing TB IPC guidelines and assess possible differences. For example, I suggest the authors to cite the study: Azeredo ACV et al. Tuberculosis in Health Care Workers and the Impact of Implementation of Hospital Infection-Control Measures. Workplace Health Saf 2020; doi:10.1177/2165079920919133.

Reviewer #2: The tool and the data used for facility assessment survey and for the observation of TB infection prevention and control practices were not included.

All other comments to the authors are in the review note.

Reviewer #3: Thank you to the authors for this important work. Please see my comments below.

Introduction

Provide statistics to illustrate the incidence of PTB and MDR TB in Bangladesh. How is TB managed in the country? The study focuses on hospitalisation TB patients. Are all TB patients hospitalised, or are they treated in the community? Where are patients diagnosed – at clinics, hospitals? It would be useful for the reader to have a better understanding of how the TB program works in Bangladesh.

Are the National Tuberculosis Control Program TB IPC guidelines based on the 2009 WHO Policy on TB Infection Control in Health-Care Facilities, Congregate Settings and Households? This should be clarified so that the reader has a better context of the Bangladesh guidelines (in the reference list there is no website indicated for the reader to see what appears in these guidelines).

Consider removing “well-known” information from the introduction. For example, how PTB is spread.

Material and method

The study sites should be explained in more detail. Why the inclusion of TB specialist hospitals and tertiary care hospitals. What are their different functions? It appears as if one of the hospitals was a non-government hospital. Please explain. The nature of the hospitals may influence the responses regarding the implementation of IPC.

Did you interview TB managers from the Ministry of Health? It would be interesting to hear their perspectives on the implementation of the national TB infection control guidelines.

This appears to be a mixed methods study – qualitative interview and quantitative observations? Please explain the type of interviews conducted. Were they semi-structured interviews? It should be clear that the facility assessment tool was an observation checklist. Over what period of time were the observations conducted (e.g. when triage, cough etiquette and patients wearing a surgical masks were observed was it once off, for a day, etc.). Where in the hospital did you look if there were posters on IPC? In the tertiary hospitals, were did you observe UVGI?

Describe the interview guide and assessment tool in more detail. Was it based on the national TB infection control guidelines? This is important to know as you were assessing the implementation of the guidelines. Consider including a table that explains the different controls that should be implemented according to the National TB infection control guidelines.

It appears as if the qualitative data was analysed thematically? Please mention this under the data analysis section.

Include the Ethics Clearance number.

Did you need to obtain permission from the Ministry of Health to undertake this study?

Results

Were all participants working directly with TB patients?

Describe the role of the project director.

It would be useful to indicate separately what the laboratory personnel did to ensure TB IPC. Their roles are somewhat different from that of nurses and doctors working directly with TB patients.

It would be useful at the start of each section to have an introductory sentence to explain the information that will follow. For example: In terms of managerial controls we looked at the availability of TB infection control committees and facility specific IPC plans.

Table 1 is confusing. It is not clear if the activities listed were all in place? Consider deleting this table.

As no one appeared to be aware of the National TB infection control guidelines, did you follow this up at a higher level? For example, with a TB manager at national level?

Why is “Operational Research” a new heading? Does it fall under “managerial controls”?

Why is the risk allowance discussed under administrative controls?

Please explain line 187-188: There was no prevention or healthcare package available for HCWs in the study hospitals.

Explain how the hospitals lacked space to put up signs for restricted areas.

How did the pathologists ensure cough etiquette during sputum collection? Was this done in a separate room, outside? Do the pathologists collect sputum? Do nurses and doctors also do this?

Natural ventilation is mentioned under administrative controls. This should be moved to the section on environmental controls.

There appears to be some overlap in the sections on managerial and administrative controls, particularly with regard to workplace policies/plans and notification/surveillance of HCWs with TB.

Line 244-246: The observation relating to N95 respirators in the tertiary care hospital – clarify if this observation was done in a TB ward.

Include more direct quotes from the interviews.

Discussion

Keep the first paragraph more general. Save the recommendations for later in the discussion.

Line 294: The third vital step… Step 1 and 2 are not specifically indicated, so it is confusing to see a third step.

Line 310-312: It is not a limitation of the study that key informants were not aware of the national guidelines (unless you interviewed the wrong key informants). This is acually an important finding. It would have been useful to follow-up on this with the relevant mangers at national level, to see why the guidelines have not been rolled out.

The main recommendations should be to ensure that the guidelines are rolled out to all facilities and HCWs should be trained on these guidelines.

References

Check for accuracy and completeness. For example:

Reference 3 – the actual website is missing.

Sometimes days, months and years are provided while in other instances only the year is provided.

The manuscript should be professionally language edited.

6. PLOS authors have the option to publish the peer review history of their article (what does this mean?). If published, this will include your full peer review and any attached files.

Reviewer #1: No

Reviewer #2: No

Reviewer #3: No

---

## [Author Response · Author response to Decision Letter 0]

17 Dec 2020

Response to editor’s comments

Comments: Journal Requirements: When submitting your revision, we need you to address these additional requirements.

and

Response: Thank you. Based on your suggestion, we reviewed the PLOS ONE’s style requirements and revised the authors’ affiliation and body of the manuscript. 

Comments: 2. We suggest you thoroughly copyedit your manuscript for language usage, spelling, and grammar. If you do not know anyone who can help you do this, you may wish to consider employing a professional scientific editing service. 

Response: Based on your suggestion, we contacted Editage who reviewed and edited our manuscript for grammar and English language expression. As recommended, we uploaded two versions of the manuscript: a clean version and a track-change version. 

Comment: 3. Under data analysis, the team reviewed the transcripts multiple times. Please specify how many times the team reviewed and if inter coder agreement was assessed.

Response: We revised the data analysis section. Now it reads, “The team transcribed all audio recorded KIIs verbatim and reviewed each transcript at least twice. Three team members who were involved in data collection coded the data. The first authors developed a code list along with code definitions. The team then reviewed the transcriptions line by line, coded the data and summarized the data under emerging and pre-defined themes based on the research questions that was aligned with the four broader TB IPC measures. When disagreement occurred, the team discussed the codes with their definitions with the senior author (MSI) to reach into consensus and thus inter coder agreement was achieved” on page 7 lines 151-157 on the clear version of the manuscript. We did not perform intercoder reliability separately. 

Comment: 4. Under study design, you indicate that the study utilized both qualitative and quantitative data - which were entered into MS excel for descriptive analysis. However, there are no texts or tables summarizing quantitative findings. If there are no quantitative findings to report, please delete the methods section to show that only qualitative data were collected.

Response: Many thanks for your comment. We updated the method section and excluded the quantitative method from the method section on page 5. 

Comment: 5. As part of your revision, please complete and submit a copy of the COREQ Guidelines checklist, a document that aims to improve experimental reporting and reproducibility of qualitative studies for purposes of post-publication data analysis and reproducibility: https://www.equator-network.org/reporting-guidelines/coreq/. Please include your completed checklist as a Supporting Information file. Note that if your paper is accepted for publication, this checklist will be published as part of your article

Response: We completed and submitted the COREQ Guidelines checklist information as a supplementary file.

Comment: 6. We note that you have indicated that data from this study are available upon request. PLOS only allows data to be available upon request if there are legal or ethical restrictions on sharing data publicly. For information on unacceptable data access restrictions, please see http://journals.plos.org/plosone/s/data-availability#loc-unacceptable-data-access-restrictions.

Response: According to icddr,b data sharing policy, data will not be available in public repositories. One copy of the complete dataset (anonymized and decoded) and metadata will be shared with the icddr,b repository team after completion of the study. Data access will be subject to the icddr,b data policy (http://www.icddrb.org/policies) upon approval from institutional review board. Interested parties may contact Ms. Armana Ahmed (aahmed@icddrb.org) with further inquiries related to data access. We added this in the cover letter.

Comments: a) If there are ethical or legal restrictions on sharing a de-identified data set, please explain them in detail (e.g., data contain potentially identifying or sensitive patient information) and who has imposed them (e.g., an ethics committee). Please also provide contact information for a data access committee, ethics committee, or other institutional body to which data requests may be sent.

Response: Please see our response above.

Response to reviewers’ comments

Reviewer #1: 

Comment: In this article the authors aimed to assess the implementation of the TB IPC healthcare measures in health settings, Bangladesh. This is an important and interesting study.

They identified poor implementation of TB IPC measures in the study settings. In the ‘Discussion’ section, the authors cited other studies that found the same problems in implementation. However, it is important to cite studies that have been successful in implementing TB IPC guidelines and assess possible differences. For example, I suggest the authors to cite the study: Azeredo ACV et al. Tuberculosis in Health Care Workers and the Impact of Implementation of Hospital Infection-Control Measures. Workplace Health Saf 2020; doi:10.1177/2165079920919133.

Response: Thank you for your valuable comment. As suggested, we reviewed the paper, found it relevant and cited it in the discussion section of the manuscript on page 20 lines 384-387 on the clear version of the manuscript.

Reviewer #2:

Comment: The tool and the data used for facility assessment survey and for the observation of TB infection prevention and control practices were not included. All other comments to the authors are in the review note.

Response: Now, we have uploaded the data collection tools as supplementary information (SI-1).

Comment: In the introduction section: Line 53: The sentence ‘TB mortality rate was 36 per 100,000’ is not complete, Mortality rate globally or locally? 

Response: We have revised and updated the information. Now, it reads, “The estimated incidence of TB per 100,000 is 221 in Bangladesh, with a mortality rate of 24 per 100,000 population (1).” on page 3 lines 49-50.

Comment: Line 58: Are all healthcare workers exposed or those working in TB clinics?

Response: “In middle- and low-income high TB-burden countries, all healthcare workers (HCWs) are at risk of TB exposure due to the presence of presumptive and or confirmed TB patients in the hospital (2-4)” is added on page 4 lines 67-68. 

Comment: In the methodology section: - It will be nice to know the number of persons that were involved in the data collection and the geographical distribution of the healthcare settings used for this study across Bangladesh. (Lines 96-99).

Response: Thank you for your query. We have updated the method section. Now it reads, “This was a qualitative study where we utilized key informant interviews (KIIs), observation, and a facility assessment checklist as data collection tools. The field team consisted of four males and one female researcher, trained in social science research with approximately five years of TB-related research experience, who collected the data. The field team had prior working relations with the study facility management teams, and this helped to build good rapport with the participants. The field team sought written permission from all the facilities before the data collection commenced. The TB specialty hospitals were situated in Dhaka, Rajshahi, Sylhet, Barishal, Chittagong, Khulna, Mymensingh, and Pabna, and the tertiary care hospitals were situated in Rajshahi, Barishal, and Kishoreganj.” on page 6 lines 108-116. 

Comment: In what language was the KII conducted?

Response: “After obtaining informed written consent, three researchers trained in social science with several years of experience in qualitative research conducted the KIIs in the Bengali language.” is added on page 6 lines 121-122. 

Comment: Ethics: The name of the IRB and the approval number should be provided. (lines 112-114)

Response: We added the name of the IRB and the approval number on page 8 lines 162-165 as, “The study protocol was reviewed and approved by the Research Review Committee and Ethical Review Committee of icddr,b (IRB number: PR#12067). The field team obtained written informed consent from the study participants. This study protocol was reviewed and approved by the NTP under the Ministry of Health and Family Welfare, government of Bangladesh.”

Comment: In the Results section: Table 2: It will be nice to know which of the distribution of the hospitals either as TB specialty hospitals or tertiary care hospitals instead of numbering.

Response: Thank you, we revised the Table-2, where now we added one column showing the type of the hospitals as TB specialty hospitals and tertiary care hospitals and removed the number column on page 14. 

Comment: How many of the participating physicians were directors of the hospitals, heads of medicine and senior physicians? (Line 117)

Response: Between February and June 2018, the team conducted 59 unstructured KIIs with hospital directors (10), heads of medicine units (five), senior physicians (eight) and junior physicians (five) of inpatient and outpatient departments, laboratory personnel (19), and nursing supervisors (11) and administrative worker (one). This was revised on page 6 lines 116-119. 

Comment: Line 128: ‘only laboratory equipment’s monitoring in terms of functionality were in place in four health settings’--- were these hospital TB specialty hospitals or tertiary care hospitals?

Response: For clarity, we revised and updated the sentence as, “Monitoring of laboratory equipment functionality was operational in only four TB specialty hospitals” on page 9 lines 187-188. 

Comment: Operational research: the category of the only centre that carried out operational research should be stated. (Line 176)

Response: Thank you. Now we added, “Regarding operational research, the respondents repeatedly mentioned that research requires a budget, planning, and manpower. One of the study settings (non-government TB specialty hospital) conducted operational research on TB control” on page 12 lines 229-231.

Comment: Line 216: How many of the TB specialty hospitals are non-governmental?

Response: Only one TB specialty hospital was non-governmental (private), added the information in the manuscript on page 5 lines 105. 

Comment: Line 228: without categorizing the hospital settings in table 2, the statement ‘All the tertiary care hospitals lacked UVGI (Table 2)’ is not justifiable.

Response: Yes, thank you for noticing it, we found that there was no UVGI in all the three tertiary-care hospitals (Table 2). We looked for UVGI lights in the emergency, outdoor, medicine wards, pathology and radiology department, this information is available on page 16 lines 298-302.

Comment: References: 

Number 7: List the names of five authors before et al.

Number 21: The journal name, issue no, volume if any and page numbers should be provided.

Response: Thank you, we now followed the PLOS one journal reference style using EndNote. 

Comment: In addition, the whole manuscript should be edited and grammatical errors addressed for example not limited to;

Response: We edited the manuscript for grammatical errors by a professional English language editing service, “Editage”.

Comment: Line 118-119: ‘The mean age of the participants was 45 years with mean job duration of 10 years’

Response: We have revised the sentence. Now it reads, “The mean age of the participants was 45 years, with a mean job duration of 10 years” on page 8 lines 174-175.

Comment: Line 313-314: ‘To enable the health system for better implementation of TB IPC guideline, the national TB IPC guidelines should be introduced in all health settings dealing with TB cases…..’ 

Response: We have revised the sentence. Now it reads, “To enable the health system to better implement the national TB IPC guidelines, the guidelines should be introduced in all health settings dealing with TB cases through participatory approaches” on page 20 lines 395-397.

Comment: Line 319: ‘guidelines among the hospitals that resulted in poor adherence to the TB IPC measures…..’

Response: We have revised the sentence. Now it reads, “Most studies, including this one, have identified poor adherence to the implementation of TB IPC guidelines among HCWs” on page 20 lines 383-384.

Reviewer #3: 

Comment: Thank you to the authors for this important work. Please see my comments below.

Introduction: Provide statistics to illustrate the incidence of PTB and MDR TB in Bangladesh. 

Response: Thank you. Based on your suggestion we revised the introduction of the manuscript and added information about the incidence PTB and MDR TB in Bangladesh. Now, it reads, 

“In 2019, 10 million people were infected with TB globally; 79% were in the 30 high-burden countries, and 1.2 million people died from TB(1) . Bangladesh is one of the 30 high TB-burden countries and accounts for 3.6% of the global total. The estimated incidence of TB per 100,000 is 221 in Bangladesh, with a mortality rate of 24 per 100,000 population (1). Approximately 80% of all TB cases in Bangladesh are pulmonary TB (5). The Global TB Report 2020 estimated that 0.7% of new cases and 11% of previously treated cases are found to be positive for multidrug-resistant TB (MDR-TB), which has an incidence rate of 2.0 per 100,000 population in Bangladesh (1) ” on page 3 lines 46-54 on the clear version of the manuscript.

Comments: How is TB managed in the country? The study focuses on hospitalisation TB patients. Are all TB patients hospitalised, or are they treated in the community? Where are patients diagnosed – at clinics, hospitals? It would be useful for the reader to have a better understanding of how the TB program works in Bangladesh.

Response: Based on your recommendation, we added, “The Bangladesh national guidelines and operational manual for TB control recommend treating TB patients in a TB hospital or Directly Observed Treatment Short-course (DOTS) clinic (6). For TB-patient treatment, DOTS therapy is considered the most effective and sustainable part of the National Tuberculosis Control Program (NTP). In hospitals, the guidelines recommend the enrollment and hospitalization of a drug-resistant TB patient or TB patient with co-morbidity in a designated TB or MDR-TB ward. Due to the high number of patients, limited number of beds, lengthy treatment procedure, and lack of patient monitoring mechanisms, the government of Bangladesh also initiated community-based programmatic management of drug-resistant TB (7). The community-based programmatic management of drug-resistant TB guidelines recommend the admission of drug-resistant TB patients in chest disease hospitals for a minimum of four weeks or until two consecutive sputum smear microscopies become negative one week apart before sending them to the community” on page 3 lines 55-66.

Comments: Are the National Tuberculosis Control Program TB IPC guidelines based on the 2009 WHO Policy on TB Infection Control in Health-Care Facilities, Congregate Settings and Households? This should be clarified so that the reader has a better context of the Bangladesh guidelines (in the reference list there is no website indicated for the reader to see what appears in these guidelines).

Response: Yes, the National Tuberculosis Control Program TB IPC guidelines were based on the 2009 WHO Policy on TB Infection Control in Health-Care Facilities, Congregate Settings and Households. We added, “Based on the WHO’s 2009 policy on TB infection control in healthcare facilities, congregate settings, and households, the NTP of the government of Bangladesh developed TB IPC guidelines (http://etoolkits.dghs.gov.bd/sites/default/files/national_guidelines_for_tuberculosis_infection_control.pd) in 2011 as a part of health system strengthening (8, 9) on page 4 lines 82. 

Comment: Consider removing “well-known” information from the introduction. For example, how PTB is spread.

Response: Thank you for your comment. As suggested, we revised the introduction and omitted the well-known information from the introduction. 

Comments: Material and method 

The study sites should be explained in more detail. Why the inclusion of TB specialist hospitals and tertiary care hospitals. What are their different functions? It appears as if one of the hospitals was a non-government hospital. Please explain. The nature of the hospitals may influence the responses regarding the implementation of IPC.

Response: Thank you for your in-depth review, we updated as, “A field team of five members, consisting of an epidemiologist (one), social scientists (two), a physicist (one), and a medical technician (one), conducted this study in 11 health settings: eight TB specialty hospitals (seven public and one private) and three tertiary care hospitals (two public and one private) in Bangladesh. The rationale for selecting these hospitals was based on the fact that TB specialty hospitals admit and treat TB patients on a regular basis, whereas tertiary care hospitals admit presumptive TB patients until diagnosis and subsequently refer confirmed TB patients to either DOTS clinics or TB specialty hospitals. These hospitals also serve the largest number of TB patients in the country. Based on our prior experience working in Bangladeshi hospitals, TB patient management and the implementation of TB IPC are likely to vary between government and non-government hospitals. Therefore, we also included one non-government hospital in our study” under the study sites on pages 5, lines 96-106.

Comments: Did you interview TB managers from the Ministry of Health? It would be interesting to hear their perspectives on the implementation of the national TB infection control guidelines.

Response: This was beyond the scope of this study and therefore, we did not interview TB managers from the ministry of health. 

Comments: This appears to be a mixed methods study – qualitative interview and quantitative observations? Please explain the type of interviews conducted. Were they semi-structured interviews? It should be clear that the facility assessment tool was an observation checklist. Over what period of time were the observations conducted (e.g. when triage, cough etiquette and patients wearing a surgical mask were observed was it once off, for a day, etc.). Where in the hospital did you look if there were posters on IPC? In the tertiary hospitals, were did you observe UVGI?

Response: Thank you. Based on your comments, we have updated the method section. Now it reads, “This was a qualitative study where we utilized key informant interviews (KIIs), observation, and a facility assessment checklist as data collection tools. The field team consisted of four male and one female researcher, trained in social science research with approximately five years of TB-related research experience, who collected the data. The field team had prior working relations with the study facility management teams, and this helped to build good rapport with the participants. The field team sought written permission from all the facilities before the data collection commenced. The TB specialty hospitals were situated in Dhaka, Rajshahi, Sylhet, Barishal, Chittagong, Khulna, Mymensingh, and Pabna, and the tertiary care hospitals were situated in Rajshahi, Barishal, and Kishoreganj. Between February and June 2018, the team conducted 59 unstructured KIIs with hospital directors (10), heads of medicine units (five), senior physicians (eight) and junior physicians (five) of inpatient and outpatient departments, laboratory personnel (19), and nursing supervisors (11) and administrative workers (one). The participants were selected purposively, and the interviewer approached the respondents face to face. After obtaining informed written consent, three researchers trained in social science with several years of experience in qualitative research conducted the KIIs in the Bengali language. Through KIIs, the team investigated the presence of a TB IPC committee or plan, surveillance and assessment of TB among HCWs, staff training, monitoring and evaluation of TB IPC, advocacy or communications for TB IPC implementation, triage and separation of TB patients, cough etiquette, and personal protective measures using respirators. All the interviews were audio-recorded, and the mean duration of the interviews was 42 min. Using an open-ended interview guide, the field team conducted the interviews. The time and venue for the interviews were selected based on the respondents’ preferences. Each day, after data collection, the field team convened and discussed the interview findings. The team continued interviewing participants until data saturation was achieved, and no new data were obtained from additional interviews. We did not conduct any repeat interviews; however, a few of the respondents were re-engaged to clarify any findings from the interviews. Based on the findings, a report was prepared and shared with all participant hospitals for review and approval. Using a facility assessment tool, the team documented the presence or absence of the following: a TB IPC coordination committee or plan, TB surveillance among HCWs, training, triage, separation/cohorting of patients with pulmonary TB, cough etiquette, ventilation, fans, ultraviolet germicidal irradiation (UVGI), and respirators available for staff and fit tests or fit checks. A team of three field researchers conducted a total of 88 h (eight hours per facility) of structured observation. The tertiary care hospitals lacked a separate ward for TB patients. Presumptive pulmonary TB patients were admitted to the adult medicine wards with other general patients. Therefore, we conducted observations in the adult medicine wards of tertiary care hospitals. During observation, the team documented the number of functional fans, UVGI lights, doors and windows (including how many of them were open), fanlights, and exhaust fans as well as the presence of air conditioning, use of N95 respirators among ward occupants, use of surgical/cloth masks, use of gloves, observance of cough etiquette, use of a triage checklist, separation of presumptive pulmonary TB patients, presence of posters, and presence of signs for restricted areas or any directional sign. The field team also looked for IPC posters in the outdoor, emergency, waiting, and entrance areas of the facilities” on pages 5-6, lines 108-149. 

Comments: Describe the interview guide and assessment tool in more detail. Was it based on the national TB infection control guidelines? This is important to know as you were assessing the implementation of the guidelines. Consider including a table that explains the different controls that should be implemented according to the National TB infection control guidelines.

Response: The interview guide and assessment tool were based on TB infection control guidelines. We also added the data collection tools as supplementary documents. We revised the table-1, according to your advice on page 10.

Comments: It appears as if the qualitative data was analyzed thematically? Please mention this under the data analysis section.

Response: Yes, the qualitative data was analyzed thematically. We have revised and updated the data analysis section as, “The team transcribed all audio recorded KIIs verbatim and reviewed each transcript at least twice. Three team members who were involved in data collection coded the data. The first authors developed a code list along with code definitions. The team then reviewed the transcriptions line by line, coded the data, and summarized the data under emerging and predefined themes based on the research questions that were aligned with the four broader TB IPC measures. When disagreements occurred, the team discussed the codes and their definitions with the senior author (MSI) to reach a consensus, and thus, intercoder agreement was achieved. The team also reviewed the facility survey data and extracted the frequency of the activities into a spreadsheet, and a descriptive analysis was performed. The facility assessment tool was adopted from the national tuberculosis infection control guidelines(8, 10)” on pages 7- 8, lines 151-160. 

Comments: Include the Ethics Clearance number.

Did you need to obtain permission from the Ministry of Health to undertake this study?

Response: We have updated the section. Now it reads, “The study protocol was reviewed and approved by the Research Review Committee and Ethical Review Committee of icddr,b (IRB number: PR#12067). The field team obtained written informed consent from the study participants. This study protocol was reviewed and approved by the NTP under the Ministry of Health and Family Welfare, government of Bangladesh” on page 8 lines 162-165. 

Comments: Results

Were all participants working directly with TB patients? Describe the role of the project director.

It would be useful to indicate separately what the laboratory personnel did to ensure TB IPC. Their roles are somewhat different from that of nurses and doctors working directly with TB patients.

Response: All the participants had exposures to TB patients or their specimens but may not on a regular basis. We added, “The project director was involved in the overall monitoring and supervision of the TB project activities as well as TB patient management in the hospital. The project director and 10 hospital directors were predominantly involved in administrative activities and implementation of policies recommended by the Ministry of Health and other partners. Physicians and nurses worked directly with TB patients, and lab workers were regularly exposed to patients and their specimens” on page 8, lines 169-174.

Comments: It would be useful at the start of each section to have an introductory sentence to explain the information that will follow. For example: In terms of managerial controls we looked at the availability of TB infection control committees and facility specific IPC plans.

Response: Thank you. We now added topic sentences in most of the paragraphs in the manuscript.

Comments: Table 1 is confusing. It is not clear if the activities listed were all in place? Consider deleting this table.

Response: We revised the table-1 to make it clearer and more communicative on page-10. 

Comments: As no one appeared to be aware of the National TB infection control guidelines, did you follow this up at a higher level? For example, with a TB manager at national level?

Response: Yes, we organized a dissemination seminar where we invited stakeholders from NTP, ministry of health, USAID, US CDC and other local partners working on TB, and presented the study findings.

Comments: Why is “Operational Research” a new heading? Does it fall under “managerial controls”?

Response: Thank you. We have deleted the paragraph heading.

Comments: Why is the risk allowance discussed under administrative controls? Please explain line 187-188: There was no prevention or healthcare package available for HCWs in the study hospitals.

Response: Thank you. We removed this to tighten our findings under the administrative control measures. 

Comments: Explain how the hospitals lacked space to put up signs for restricted areas.

Response: Thank you for identifying this mistake. We revised the sentence as, “All hospitals lacked clear signs, such as “restricted area” signs, directional signage, or hospital guidance signage to assist people and keep them away from restricted areas” on page 13 lines 257-259. We removed the word “space”. 

Comments: How did the pathologists ensure cough etiquette during sputum collection? Was this done in a separate room, outside? Do the pathologists collect sputum? Do nurses and doctors also do this?

Response: Thank you. We added the findings, “Three public and one private TB specialty hospital collected sputum outdoors (outside buildings, in an open place). The three public TB specialty hospitals collected sputum in the outdoor corridor, whereas three tertiary care hospitals and one public TB specialty hospital collected sputum in a designated indoor area. All hospitals lacked clear signs, such as “restricted area” signs, directional signage, or hospital guidance signage to assist people and keep them away from restricted areas. Typically, the pathologists would not collect sputum samples themselves; rather, they trained laboratory assistants to collect sputum and concurrently enforce the maintenance of cough etiquette by patients (patients were instructed on cough etiquette before sputum induction procedures). The laboratory assistant safely disposed of the sputum cup and sticks in the dedicated waste disposal container. Hence, duty doctors or nurses were not involved in sputum collection” on page 13, lines 253-264. 

Comments: Natural ventilation is mentioned under administrative controls. This should be moved to the section on environmental controls.

Response: Thank you for noticing it, we have moved it to environmental section now, on page 16.

Comments: There appears to be some overlap in the sections on managerial and administrative controls, particularly with regard to workplace policies/plans and notification/surveillance of HCWs with TB.

Response: Yes, we mistakenly placed the screening part under the managerial control, now we have moved it under the administrative control on page 13.

Comments: Line 244-246: The observation relating to N95 respirators in the tertiary care hospital – clarify if this observation was done in a TB ward. Include more direct quotes from the interviews.

Response: Under the method section, we added, “The tertiary care hospitals lacked a separate ward for TB patients. Presumptive pulmonary TB patients were admitted to the adult medicine wards with other general patients. Therefore, we conducted observations in the adult medicine wards of tertiary care hospitals” on page 7, lines 140-142. We also added one direct quote, “We seldom wear N95 respirators, and I cannot remember when I last used N95 respirators while attending a pulmonary TB patient” on page 17, lines 323-324.

Comments: Discussion

Keep the first paragraph more general. Save the recommendations for later in the discussion.

Response: Based on your suggestion, we moved the recommendations at the end of the discussion section on page 20, lines 395-404.

Comments: Line 294: The third vital step… Step 1 and 2 are not specifically indicated, so it is confusing to see a third step.

Response: Yes, we understand our error here. We removed the word “third vital step”. 

Comments: Line 310-312: It is not a limitation of the study that key informants were not aware of the national guidelines (unless you interviewed the wrong key informants). This is actually an important finding. It would have been useful to follow-up on this with the relevant mangers at national level, to see why the guidelines have not been rolled out.

Response: Thank you very much for this comment. We have revised the limitation section as, “This study has several limitations. Most of the key informants were senior HCWs. These senior management employees may not have free time to participate in TB IPC training; therefore, they remain unaware of the TB IPC activities. Second, this study was conducted in 11 hospitals that were not randomly selected and therefore, the findings may not be generalizable to all hospitals that provide care to TB patients. However, this study findings are consistent with prior reports and studies conducted in other government and non-government hospitals in Bangladesh (4, 11, 12)” on page 20 lines 388-393. 

 We are also in touch with the NTP to work on implementation of TB IPC in Bangladeshi hospitals.

Comments: The main recommendations should be to ensure that the guidelines are rolled out to all facilities and HCWs should be trained on these guidelines.

Response: Thank you for valuable suggestion. We added this recommendation, “The national TB infection control guidelines should be rolled out in chest-disease hospitals, clinics, and tertiary care hospitals. Frontline HCWs should be trained on different IPC measures outlined in the national TB infection control guidelines” on page 20 lines 397-401.

Comments: References

Check for accuracy and completeness. For example:

Reference 3 – the actual website is missing.

Sometimes days, months and years are provided while in other instances only the year is provided.

Response: We revised the references accordingly.

Comments: The manuscript should be professionally language edited.

Response: The manuscript has been reviewed and edited by a professional language editing service-Editage. 

Reference

1. World Health Organization. Global Tuberculosis Report. Geneva: World Health Organization; 2020.

2. Joshi R, Reingold AL, Menzies D, Pai M. Tuberculosis among health-care workers in low-and middle-income countries: a systematic review. PLoS Med. 2006;3(12):e494.

3. Nasreen S, Shokoohi M, Malvankar-Mehta MS. Prevalence of latent tuberculosis among health care workers in high burden countries: a systematic review and meta-analysis. PloS one. 2016;11(10):e0164034.

4. Islam MS, Chughtai AA, Nazneen A, Chowdhury KIA, Islam MT, Tarannum S, et al. A tuberculin skin test survey among healthcare workers in two public tertiary care hospitals in Bangladesh. PLoS One. 2020;In Press.

5. World Health Organization. Global Tuberculosis Report. Geneva: World Health Organization; 2018.

6. National Tuberculosis Control Programme. National guidelines and operational manual for tuberculosis control Dhaka, Bangladesh: USAID, National Tuberculosis Control Programme, World Health Organization; 2015.

7. National Tuberculosis Control Programme. National Guidelines and Operational Manual for Programmatic Management of Drug Resistant TB (PMDT). Dhaka, Bangladesh: National Tuberculosis Control Programme, World Health Organization Country Office for Bangladesh; 2013.

8. National Tuberculosis Control Programme. National Guidelines for Tuberculosis Infection Control. Dhaka, Bangladesh: National Tuberculosis Control Programme, USAID. TB CARE II, Bangladesh, World Health Organization; 2011.

9. World Health Organization. WHO policy on TB infection control in healthcare facilitis, congregate settings and households. WHO,Geneva: WHO/HTM/TB/2009; 2009.

10. Ahmed FW. A critical analysis of Bangladesh national tuberculosis control program Journal of Pulmonology and Clinical Research 2018;2(1):16-9.

11. Rimi NA, Sultana R, Luby SP, Islam MS, Uddin M, Hossain MJ, et al. Infrastructure and contamination of the physical environment in three Bangladeshi hospitals: putting infection control into context. PloS one. 2014;9(2):e89085.

12. Rahman MS, Ayub A, Rahman L, Saki NA, Khan MH, Faisel AJ, et al. Institutionalizing Infection Prevention and Control in a TB and Lung Disease Hospital in Bangladesh. Bangladesh: USAID, IRD, MSH, KNCV, Challenge TB; 2019.

---

## [Editor Report · Decision Letter 1]

26 Jan 2021

PONE-D-20-22170R1

Implementation status of national tuberculosis infection control guidelines in Bangladeshi hospitals

PLOS ONE

Dear Dr. Nazneen,

Thank you for submitting your manuscript to PLOS ONE. After careful consideration, we feel that it has merit but does not fully meet PLOS ONE’s publication criteria as it currently stands. Therefore, we invite you to submit a revised version of the manuscript that addresses this point raised during the review process.

While your study is largely qualitative, the observation checklist completed at 11 facilities constitutes a quantitative component. Please consider revising this under the methods section. 

We look forward to receiving your revised manuscript.

Kind regards,

Michelle Engelbrecht, PhD.

Academic Editor

PLOS ONE

Additional Editor Comments (if provided):

Thank you for addressing the reviewers' comments so thoroughly. The article reads well. Just one additional comment, while your study was mainly qualitative, it did contain a quantitative component, namely the observation checklist that was completed at 11 hospitals. Consider revising accordingly in the methods section.

---

## [Author Response · Author response to Decision Letter 1]

27 Jan 2021

Response to editor’s comment

Comments: While your study is largely qualitative, the observation checklist completed at 11 facilities constitutes a quantitative component. Please consider revising this under the methods section. 

Response: Thank you. Based on your suggestion, we have revised the method section in the abstract and in the manuscript. In the abstract, we added, “Between February and June 2018, we conducted a mixed-method study at 11 health settings…... on page 2”. In the method section, we added, “This was a mixed-method study. We used both qualitative and quantitative data collection tools that included key informant interviews (KIIs), observation, and a facility assessment checklist” on page 5.

---

## [Editor Report · Decision Letter 2]

29 Jan 2021

Implementation status of national tuberculosis infection control guidelines in Bangladeshi hospitals

PONE-D-20-22170R2

Dear Dr. Nazeen

We’re pleased to inform you that your manuscript has been judged scientifically suitable for publication and will be formally accepted for publication once it meets all outstanding technical requirements.

Kind regards,

Michelle Engelbrecht, PhD.

Guest Editor

PLOS ONE
---

## [Editor Report · Acceptance letter]

3 Feb 2021

PONE-D-20-22170R2 

Implementation status of national tuberculosis infection control guidelines in Bangladeshi hospitals 

Dear Dr. Nazneen:

I'm pleased to inform you that your manuscript has been deemed suitable for publication in PLOS ONE. Congratulations! Your manuscript is now with our production department. 

Kind regards, 

on behalf of

Dr. Michelle Engelbrecht 

Guest Editor

PLOS ONE